# Perceived fairness of claimants undergoing a work disability evaluation: Development and validation of the Basel Fairness Questionnaire

**Regine Lohss[1], Timm Rosburg[1]\*, Monica Bachmann[1], Brigitte Walter Meyer[1], Wout de Boer[1], Katrin Fischer[2], Regina Kunz[1]**

1 Department of Clinical Research, EbIM, Research & Education, University Hospital Basel, Basel, Switzerland, 2 Institute Humans in Complex Systems, School of Applied Psychology, University of Applied Sciences and Arts Northwestern Switzerland, Olten, Switzerland

\* timm.rosburg@usb.ch

## Abstract

### Background

There are currently no tools for assessing claimants' perceived fairness in work disability evaluations. In our study, we describe the development and validation of a questionnaire for this purpose.

### Method

In cooperation with subject-matter experts of Swiss insurance medicine, we developed the 30-item Basel Fairness Questionnaire (BFQ). Claimants anonymously answered the questionnaire immediately after their disability evaluation, still unaware about its outcome. For each item, there were four response options, ranging from "strongly disagree" to "strongly agree". The construct validity of the BFQ was assessed by running a principal component analysis (PCA).

### Results

In 4% of the questionnaires, the claimants' perception on the disability evaluation was negative (below the median of the scale). The PCA of the items responses followed by an orthogonal rotation revealed four factors, namely (1) Interviewing Skills, (2) Rapport, (3) Transparency, and (4) Case Familiarity, explaining 63.5% of the total variance.

### Discussion

The ratings presumably have some positive bias by sample selection and response bias. The PCA factors corresponded to dimensions that subject-matter experts had beforehand identified as relevant. However, all item ratings were highly intercorrelated, which suggests that the presumed underlying dimensions are not independent.

**Data Availability Statement:** All relevant data are available from Figshare (doi: 10.6084/m9.figshare.12600410.v1).

**Funding:** The study was funded by the Swiss National Accident Insurance Fund, Suva, Lucerne, granted to RK. The funder had no role in study design, data collection and analysis, decision to publish, or preparation of the manuscript.

**Competing interests:** RK reports a grant from the Swiss National Accident Insurance Fund, Suva for conducting the study. RK took a part-time position at Suva as head of the Competence Center Insurance Medicine while the study was running (01.12.2015 – 31.07.2017). The employment is still ongoing. No Suva claimant participated in the study. This does not alter our adherence to PLOS ONE policies on sharing data and materials.

## Conclusion

The BFQ represents the first self-administered instrument for measuring claimants' perceived fairness of work disability evaluations, allowing the assessment of informational, procedural, and interactive justice from the perspective of claimants. In cooperation with Swiss assessment centres, we plan to implement a refined version of the BFQ as feedback instrument in work disability evaluations.

## Introduction

In most European countries, individuals with social security coverage who consider themselves as unable to work because of poor health can file a claim for disability benefits. In Switzerland, they have to file this claim at the disability insurance (DI), via DI offices ("*IV-Stellen*") located in each canton. The DI offices seek to establish the claimant's degree of work incapacity, as critical variable determining the amount of work disability benefits. To this end, DI offices commission mono-, bi-, or multidisciplinary medical evaluations, depending on the complexity of the claimant's medical history. Mono- and bidisciplinary evaluations are directly assigned to medical experts, whereas a random procedure allocates multidisciplinary medical evaluations to assessment centres, which need to be licenced by the Swiss Federal Social Insurance Office ("*Bundesamt für Sozialversicherung*"). The assessment centres forward the claims to contracted medical experts who, based on medical records, by interviewing the patients, and by examining their health status, assess the claimants' functional capacities with regard to demands at work [1].

The legal system expects that the assigned medical experts evaluate all medical and functional aspects neutrally, objectively, and equally [2]. Moreover, the legal system expects that medical experts arrive to comparable evaluations for similar cases. Unfortunately, in practice, the interrater agreement is limited even for the very same patients [3], and the variability across experts, in particular for the assessment of mental disorders, can be surprisingly large [4]. The lack of comparability cannot easily be dissolved, due to the complexity and uniqueness of the cases. This is particularly true for the evaluation of limitations in performing work activities and of participation restrictions due to mental disorders. In 2018, 47% of Swiss work disability beneficiaries received their pension in consequence of mental disorders [5].

For claimants, the medical disability evaluations maximally consist of a few encounters with medical specialists. Rather than low agreement between medical experts in the disability evaluation, the claimant himself is more likely to experience a lack of agreement between the evaluation of the medical expert and the view of the attending general physician (or medical specialist), who might have encouraged the patient to file a claim for disability benefits. Moreover, the claimant's self-perception might diverge from the expert's assessment [6]. This discordance may lead to dissatisfaction of claimants when the expert considers the health problems as less severe than the claimant, the disability is not approved or to a lesser degree than the claimant has expected, and the disability pension / financial compensation is less than expected. In this case, claimants might consider the outcome of the evaluation and the financial compensation as unjust [7].

However, claimants might also experience the interaction with the expert as cold, disrespectful, or even demeaning, they might feel badly informed, or might find their point of view poorly considered in the evaluation. Currently, there are few feedback loops in the process of work disability evaluations to allow an assessment of these quality aspects, referring to

interactional, informational, and procedural justice [7]. Due to the lack of standardized and systematic assessment, it is at present unknown, how often, to what extent, and in what regard claimants might perceive work disability evaluations as unfair. The systematic assessment of perceived fairness in work disability evaluations is hampered not just by lacking implementation of feedback instruments, but also by the lack of instruments designed for that very purpose.

Perceived fairness might not only be considered as an important quality characteristic of work disability evaluations, but research shows that perceived unfairness also tends to worsen mental health problems of already vulnerable patients/claimants. When claiming compensation after road and workplace injuries, perceived injustice modulated clinical symptoms in claimants, with higher degrees of perceived injustice associated with higher degrees of depression, more severe pain, and poorer mental health later on [8–11]. Therefore, it appears mandatory to implement quality control measures in disability evaluations that allow the assessment of perceived fairness. Such measures would provide feedback to the medical experts about how claimants perceived their interaction during the work disability evaluation, allowing the medical experts to monitor and to adjust their interviewing practice. Moreover, the measures allow to document that the evaluations took place in compliance with certain quality standards. The aim of our study was to develop and validate a questionnaire on perceived fairness in work disability evaluations as an initial step for implementing this kind of quality control in work disability evaluations.

## Methods

### Overview of the study procedure

The study started with the collection and creation of items for evaluating patient satisfaction in disability evaluations. The item pool was repeatedly revised by feedback from subject-matter experts and pre-testing. The to-be-validated version of the questionnaire contained 30 items. Claimants answered this questionnaire, as well as 11 items from the *Cologne Patient Questionnaire* (CPQ) [12] and *Satisfaction with Life Scale* (SWLS) [13] for assessing the convergent and divergent validity. In addition, the claimants rated the overall quality of the evaluation. The assigned medical experts provided analogue ratings from their perspective ('expert questionnaire'). We ran a principal component analysis (PCA) on the claimant responses in the questionnaire to reveal its underlying factor structure. Moreover, we correlated the questionnaire's sum scores and factor loadings with scores from the other scales and ratings. The procedural steps are detailed in the following.

### Development of the Basel Fairness Questionnaire (BFQ)

Since a PubMed search did not reveal any instruments that measure perceived fairness of disability evaluations, we contacted key informants of insurance medicine to identify such instruments. We found an unpublished Dutch instrument for the assessment of patient satisfaction with regard to the experts' communication, but no instrument covering all major aspects of perceived fairness. Therefore, we opted for developing such a questionnaire and used the Dutch instrument as a starting point.

The Dutch instrument conformed to the context of disability evaluations in the Netherlands and contained multiple references to national specifics [14, 15]. We selected several items of its subscales Listening, Empathizing, Correctness, Clearness, Rigour, and Professionalism as raw material to create an item pool. Following its translation to German, six subject-matter experts in work disability evaluations (including two of the authors, WdB and RK) iteratively re-worded the items for better comprehensibility, adapted the items to the setting of Swiss

insurance medicine, discarded irrelevant items, and added items related to aspects not covered by the Dutch questionnaire. All six subject-matter experts had at minimum 15 years of practical work experience in field of insurance medicine and disability evaluations. A prefinal set of 29 items was tested for clarity, comprehensibility, and coverage of all relevant issues by applying them to more than 40 patients [16], using the thinking aloud method [17]. The pre-testing led to the deletion of five items and addition of six new items.

The final questionnaire, as used in this study, included 30 items with four response options per item, ranging from "strongly disagree" to "strongly agree", which were numerically coded from "1" to "4", respectively. Four items referring to the knowledge of the expert about the claimants' medical records had a fifth response option ("Can't tell"). Ratings on the overall perceived fairness of the evaluation and the general satisfaction with the evaluation complemented the questionnaire. These two ratings were made on a 7-point Likert scale with two anchor points: "very unfair/dissatisfied" (= 1) and "very fair/satisfied" (= 7). Moreover, the claimant could provide verbatim comments on the satisfaction with the disability evaluation, which have previously been reported [18]. Four subject-matter experts (including one author, WbB) individually grouped the 30 items into two to five thematic clusters, discussed the clusters, and iteratively reorganised the structure until they reached consensus. The final, conjoint clustering contained four clusters or dimensions, which the subject-matter experts deemed to reflect perceived fairness in disability evaluations. These dimensions were a) the interviewing skills of the expert, b) rapport (atmosphere of trust and respect), c) case familiarity (the expert's knowledge about the patient and his/her records), and d) transparency (provision of information by the expert).

In order to establish that our questionnaire is related to other instruments measuring similar constructs and shows convergent validity, we added two scales of the *CPQ* [12]: a) "patients' trust in the attending physicians" (five items) and b) "bullying by the attending physicians" (six items). The CPQ was developed to quantify patient satisfaction with all major aspects of service quality in hospital care and contains many scales of no relevance for disability evaluations. The CPQ items used for the current study were re-worded to adopt them to the context of disability evaluations. One CPQ item was identical to an item of the questionnaire ("the expert let me finish speaking"). CPQ items had four response options, ranging from "strongly disagree" to "strongly agree", which were numerically coded from "1" to "4". In order to establish that our questionnaire discriminates between the patients' satisfaction with their disability evaluation and their satisfaction with life in general (divergent validity), we included the *SWLS*, a validated instrument for assessing life satisfaction [13]. The *SWLS* contains five items with five response options per item, ranging from "strongly disagree" (= 1) to "strongly agree" (= 5). Finally, in order to check to what degree patients and medical experts agree in their judgment of the evaluation process, experts rated on 7-point Likert scales to what extent the patient presumably had perceived the evaluation as overall fair, from "very unfair" (= 1) to "very fair" (= 7). Moreover, the medical experts rated from their perspective the quality of the interaction with the patient and how well the survey of the patient succeeded during the interview from "very poor" (= 1) to very good (= 7).

## Participants

All participants were required to undergo a multidisciplinary medical evaluation for work capacity, to have sufficient command of the German language for filling out the questionnaire, and to be between 18 and 65 years old. For claimants, the participation in such medical evaluations is obligatory, whereas the participation in our study was voluntary. For the study, we recruited claimants at four Swiss assessment centres from April 2015 to March 2017, as well as

the assigned medical experts who evaluated the work capacity of the claimants. Case managers of the assessment centres (two centres in Basel, one centre in Lucerne and Binningen each) checked the eligibility of the patients for the study and sent the study information and questionnaire to potential study participants and the expert questionnaire to one of the assigned experts. Each patient assessed the perceived fairness in only one of their multidisciplinary evaluations, which was in advance specified by the assessment centre. Patients had to complete the questionnaire promptly after their evaluation while unaware of its outcome and returned the questionnaire in a sealed envelope to the study centre (EbIM). The questionnaire also asked for some basic demographic data (age, sex, marital status, profession), as well as for the medical discipline in which the work disability evaluation took place. The research team had no access to clinical records of participants. The questionnaires of experts and patients had a joint number code for linking the cases. Otherwise, the responses were anonymous. The expert questionnaire did not ask for any information on the expert in order to ensure the anonymization of the expert data. The final patient sample consisted of 305 patients (187 female, 115 male, 3 undisclosed) with a mean age of 47.4 years (SD 10.9 years). For 293 of these patients, the assigned experts rated the evaluation process and returned the questionnaires immediately after the evaluation. Cases with expert ratings but no claimant ratings were not considered for the data analysis. The medical experts were informed about the research project and were aware that the claimants would rate their performance, but they did not know the individual BFQ items.

## Statistics

For the current study purpose, we analysed the ratings across participants (i.e. we did not differentiate between the disciplines of the experts who performed the evaluations). The item difficulty and item discrimination were extracted as item characteristics: Item difficulty is defined as the ratio between the mean score of all participants and maximally achievable score [19]. Thus, the higher the value, the more respondents agreed with the item. Item discrimination is defined as the correlation between the item's scores and the total test scores. The higher the value, the higher is the degree to which an item and the whole test measure the same content [19]. Moreover, in order to assess the internal consistency of the Basel Fairness Questionnaire (BFQ) we calculated Cronbach's alpha. Cronbach's alpha describes to what extent the items of a questionnaire correlate to each other.

A PCA was run in order to determine the underlying latent variable structure of the BFQ and to compare this factor structure with the dimensions of patient satisfaction in disability evaluations as identified by subject-matter experts. We used the Kaiser-Meyer-Olkin measure of sampling adequacy and the Bartlett's test to verify that the data were suited for factor analysis. A Kaiser-Meyer-Olkin value close to 1 indicates that patterns of correlations are sufficiently independent for factor analysis to yield distinct and reliable factors [20]. A significant Bartlett's test confirms that by items relating to each other to some degree factor analysis is appropriate. The scores of the 30 items were entered into the PCA, with missing values replaced by mean scores. The number of factors was determined based on the Kaiser criterion, meaning extracted factors of the unrotated PCA needed to have an eigenvalue >1. The PCA solution was orthogonally rotated (varimax) in order to achieve a simple structure with items loading high on individual factors, which allows an easier factor interpretation.

PCA factor loadings after rotation were extracted and, together with the mean score of the questionnaire, correlated across participants with the sum scores of the two subscales of the *CPQ* (Patients' trust and Being bullied) and the sum score of *SWLS*, by calculating Pearson correlation coefficients. Likewise, the expert rating scores and global rating scores of the

patients were correlated with the rotated PCA factor loadings and the BFQ mean score. In order to have uniform polarities across all items/scores, we inverted the scores of the CPQ subscale Being bullied, as its items were negatively worded. Thus, after inversion, low values in this subscale indicated the presence of negative feelings.

### Ethical approval

The Ethics Committee Northwest/Central Switzerland approved the study in February 2014 (EKNZ: 2014–050). The claimants agreed to participate by completing the questionnaire.

### Results

Of 538 questionnaires sent to claimants in disability evaluations, 305 questionnaires (56.7%) were returned. The evaluations took place in various disciplines (psychiatry: n = 78; neurology: n = 28; rheumatology: n = 32; internal medicine: n = 40; other disciplines: n = 41; missing or ambiguous responses: n = 86). For the current study purpose, we did not differentiate between disciplines.

Across all participants, the mean scoring of the 30 items was 3.41 (SD 0.51), meaning that the feedback provided by the questionnaire was on average quite positive. The distribution was strongly left-skewed (**Fig 1**). Only 12 patients provided mean ratings below the median of the response scale (i.e. < 2.5). The on average highly positive feedback was also reflected in the

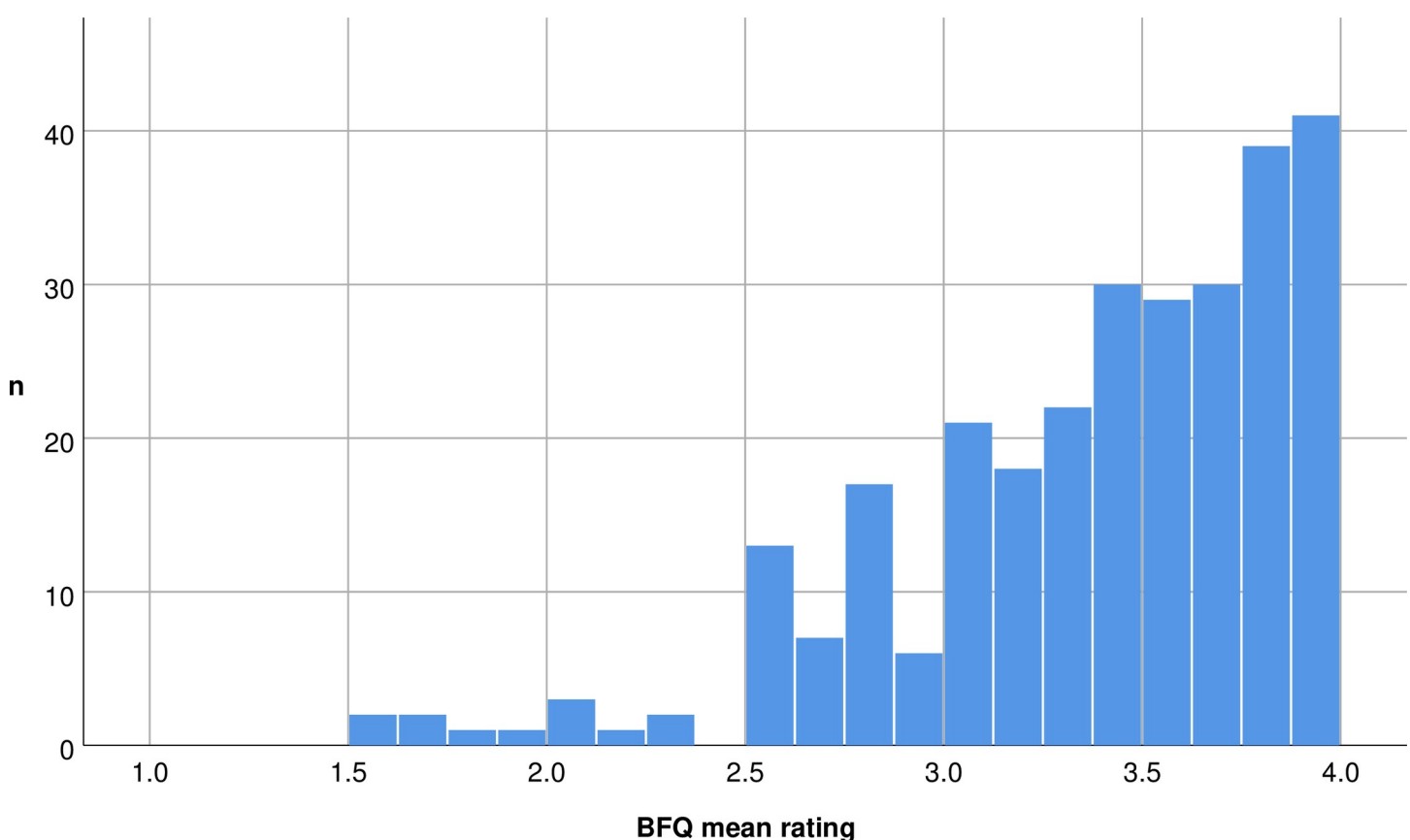

**Fig 1. The strongly left-skewed distribution of the mean BFQ item ratings across participants: The mean item rating across participants was 3.41 (SD 0.51); the individual data can be retrieved from Figs 2 to 4.**

high item difficulty (= high agreeance with item's statement, Table 1), as well as in the high ratings for the perceived fairness of the evaluation and for the general satisfaction with the evaluation, as obtained by the two separate ratings on a 7-point Likert scale. The latter ratings were 5.74 (SD 1.24) and 5.66 (SD 1.29) respectively, with a mode of "6" for both ratings. Thus, participants did not choose the most positive category ("7") when the rating scales were finer graded. However, also for these ratings, only 16 participants tended to perceive the evaluation as unfair and provided ratings below the middle of the response scale for perceived fairness (i.e. ratings < 4).

Missing responses were scarce for most items, except for the four items with the "can't tell" response option (items 2, 14, 24, and 27; Table 1). The mean ratings of participants who answered all items with four response options and of participants who produced any missing data in the latter items systematically varied (complete questionnaire group: n = 221, M = 3.52, SD = 0.46; missing data group: n = 84, M = 3.19, SD = 0.56; $F_{1, 304}$ = 27.001, p < 0.001, d = 0.668; the items with the "can't tell" response option were excluded for calculating these

**Table 1. Characteristics of the Basel Fairness Questionnaire (BFQ) items.**

| No. | Item | Miss* [n] | Item difficulty | Item discrimination | Communality |
|---|---|---|---|---|---|
| 1 | The expert took time for me. | 2 | 0.904 | 0.707 | 0.620 |
| 2 | The expert had read the records of my physicians and therapists. | 69 | 0.814 | 0.636 | 0.710 |
| 3 | The expert listened to me. | 3 | 0.894 | 0.691 | 0.615 |
| 4 | The expert went into details I told him. | 6 | 0.850 | 0.753 | 0.692 |
| 5 | The expert inquired exactly about my complaints. | 7 | 0.885 | 0.683 | 0.555 |
| 6 | I felt understood by the expert. | 6 | 0.816 | 0.823 | 0.734 |
| 7 | The expert informed me well. | 7 | 0.799 | 0.712 | 0.549 |
| 8 | I was under the impression that the expert knew how my complaints restricted me at work. | 9 | 0.664 | 0.745 | 0.618 |
| 9 | I felt taken seriously by the expert. | 10 | 0.844 | 0.807 | 0.743 |
| 10 | The expert took my complaints and restrictions seriously. | 8 | 0.813 | 0.803 | 0.655 |
| 11 | The expert responded to me in conversation. | 5 | 0.840 | 0.802 | 0.687 |
| 12 | The expert asked me how I feel. | 7 | 0.815 | 0.717 | 0.527 |
| 13 | The expert valued me. | 10 | 0.866 | 0.778 | 0.674 |
| 14 | The expert knew my medical history. | 78 | 0.725 | 0.671 | 0.746 |
| 15 | The expert treated me respectfully. | 2 | 0.905 | 0.734 | 0.652 |
| 16 | I could ask questions. | 7 | 0.862 | 0.721 | 0.521 |
| 17 | The expert was well informed about my situation. | 13 | 0.680 | 0.742 | 0.748 |
| 18 | The expert informed me about the procedure of the evaluation. | 9 | 0.732 | 0.711 | 0.657 |
| 19 | The expert informed me that she/he had to check what kind of work I was still able to do | 8 | 0.618 | 0.607 | 0.700 |
| 20 | The expert informed me about the next steps following the evaluation | 10 | 0.649 | 0.662 | 0.659 |
| 21 | The expert let me finish speaking | 6 | 0.883 | 0.651 | 0.630 |
| 22 | I was able to say everything important | 5 | 0.856 | 0.742 | 0.649 |
| 23 | The expert was empathic. | 12 | 0.826 | 0.795 | 0.690 |
| 24 | The expert paid attention to my complaints and restrictions during the evaluation. | 57 | 0.801 | 0.740 | 0.516 |
| 25 | The expert looked at me during the evaluation. | 2 | 0.900 | 0.641 | 0.551 |
| 26 | The expert asked me what I am able and not able to do. | 7 | 0.784 | 0.620 | 0.487 |
| 27 | I had the impression the expert understood my case history. | 73 | 0.812 | 0.768 | 0.613 |
| 28 | The expert involved me in the conservation. | 7 | 0.853 | 0.727 | 0.609 |
| 29 | The expert explained to me why she/he was doing something. | 10 | 0.766 | 0.645 | 0.612 |
| 30 | The expert could deal with my emotions. | 10 | 0.788 | 0.784 | 0.624 |

* Misses were defined as items with no response or "can't tell" response; only the items 2, 14, 24, and 27 had the latter response option.

mean ratings). In other words, claimants who answered all items were more positive about the evaluation than claimants who skipped items.

## Principal component analysis

Cronbach's alpha was > 0.9 ($\alpha$ = 0.969 for 30 items, n = 144; $\alpha$ = 0.961 for 26 items, n = 239). The Kaiser-Meyer-Olkin statistic was 0.957 and Bartlett's test was highly significant (p<0.001), indicating that the PCA was deemed appropriate for the data. Four factors of the unrotated PCA had eigenvalues > 1. These four factors explained 63.5% of the variance. Before rotation, a single factor dominated the factor solution, explaining 49.7% of the variance. The communalities of each item were between 0.487 and 0.748 (**Table 1**). The varimax factor rotation resulted in a so-called simple structure (high loadings on one factor and low loadings on the others), with all but two items showing loadings > 0.5 on one of the four factors (**Table 2**). Taking the content of the items into consideration, the rotated factors widely corresponded to the dimensions, as beforehand identified by the subject-matter experts: Factor 1 corresponded to Rapport, factor 2 to Interviewing Skills, factor 3 to Transparency, and factor 4 to Case Familiarity (**Table 2**).

In an exploratory analysis, the PCA was restricted to items with high loadings in factors 1 and 2, as the high loadings in one factor were mostly accompanied by moderate loadings in the other factor. For this exploratory PCA, the number of extracted factors were restricted to two. Visual inspection of the factor loadings showed that the two underlying factors were likely non-orthogonal (the factor axes did not run through chunks of items). If such a PCA was followed by an oblique (oblimin) rotation, the two extracted factors were considerably correlated (r = 0.775). Factor 1 again corresponded to Rapport, whereas factor 2 had particular high loadings on the items 21, 22, 25, and 28 (**S1 Table**). Thus, factor 2 might reflect the impression of the claimant of being part of the conversation and not just a (valued) respondent to questions. Therefore, this factor might be better understood as factor reflecting the feeling of participation (as created by the expert's interviewing skills).

Aside from items 12 and 16 (which did not load on any factor > 0.5), we identified some items, which could be removed from the questionnaire without noteworthy information loss: Some item pairs showed high correlations (>0.7), namely the items 4 and 11, 6 and 9, 9 and 10, 13 and 15, as well as 14 and 17. Taking also the semantic similarity of the items, as well as the item characteristics into account, we identified the items 9, 11, and 15 as removable items. These three items were excluded when calculating the mean ratings of each factor, as described in the following.

## Rating profile

The mean ratings of each factor were obtained by averaging the scores of the five items with the highest factor loadings in each factor. These were items 1, 3, 4, 6 and 13 for factor 1 (Rapport), items 21, 22, 23, 25, and 28 for factor2 (Interviewing Skills), items 18, 19, 20, 26, and 29 for factor 3 (Transparency), and items 2, 8, 14, 17, and 27 for factor 4 (Case Familiarity, **Tables 1** and **2**). The mean factor ratings (not to be confused with the factor loadings) were compared by a repeated measure ANOVA between the four factors. This ANOVA revealed that the mean ratings varied between the factors (F $_{3,906}$ = 155.926, p < 0.001, $\eta_p^2$ = 0.341). The mean ratings were higher for factors 1 and 2 than for factors 3 and 4, with no difference between the first two items and the latter two items (factor 1: 3.59 SD 0.52; factor 2: 3.58 SD 0.51; factor 3: 3.12 SD 0.71; factor 4: 3.16 SD 0.64). Thus, the items of factors 1 and 2 had a significantly greater item difficulty than items of factors 3 and 4.

**Table 2. Rotated factor solution.**

| No. | Item | Component | | | |
|---|---|---|---|---|---|
| | | 1 | 2 | 3 | 4 |
| 9 | I felt taken seriously by the expert. | **.736** | .307 | .162 | .283 |
| 4 | The expert went into details I told him. | **.719** | .324 | .258 | |
| 3 | The expert listened to me. | **.702** | .292 | .158 | .109 |
| 6 | I felt understood by the expert. | **.697** | .339 | .218 | .294 |
| 1 | The expert took time for me. | **.696** | .181 | .148 | .284 |
| 15 | The expert treated me respectfully. | **.672** | .392 | .158 | .145 |
| 13 | The expert valued me. | **.666** | .402 | .178 | .192 |
| 11 | The expert responded to me in conversation. | **.652** | .355 | .298 | .220 |
| 5 | The expert inquired exactly about my complaints. | **.641** | .124 | .230 | .276 |
| 10 | The expert took my complaints and restrictions seriously. | **.635** | .256 | .266 | .340 |
| 7 | The expert informed me well. | **.526** | .116 | .362 | .357 |
| 12 | The expert asked me how I feel. | .478 | .296 | .425 | .171 |
| 16 | I could ask questions. | .478 | .331 | .405 | .137 |
| 21 | The expert let me finish speaking. | .341 | **.700** | .145 | |
| 22 | I was able to say everything important. | .326 | **.631** | .354 | .142 |
| 25 | The expert looked at me during the evaluation. | .349 | **.627** | .148 | .116 |
| 23 | The expert was empathic. | .486 | **.607** | .178 | .232 |
| 28 | The expert involved me in the conservation. | .274 | **.595** | .326 | .272 |
| 24 | The expert paid attention to my complaints and restrictions during the evaluation. | .321 | **.546** | .203 | .271 |
| 30 | The expert could deal with my emotions. | .402 | **.533** | .332 | .260 |
| 19 | The expert informed me that she/he had to check what kind of work I was still able to do. | .154 | .113 | **.797** | .168 |
| 20 | The expert informed me about the next steps following the evaluation. | .262 | .165 | **.734** | .157 |
| 18 | The expert informed me about the procedure of the evaluation. | .379 | .177 | **.679** | .143 |
| 29 | The expert explained to me why she/he was doing something. | .136 | .434 | **.625** | .118 |
| 26 | The expert asked me what I am able and not able to do. | .141 | .391 | **.524** | .199 |
| 14 | The expert knew my medical history. | .259 | | .176 | **.802** |
| 2 | The expert had read the records of my physicians and therapists. | .194 | .210 | | **.790** |
| 17 | The expert was well informed about my situation. | .350 | .207 | .314 | **.696** |
| 27 | I had the impression the expert understood my case history. | .156 | .500 | .217 | **.540** |
| 8 | I was under the impression that the expert knew how my complaints restricted me at work. | .394 | .232 | .388 | **.508** |

Coefficients < 0.1 are suppressed.

## Convergent and divergent validity

304 participants answered the items of the scale Patients' trust and 303 patients the items of the scale Being bullied of the *CPQ*. The mean ratings were, similarly to the fairness questionnaire, quite high (Patients' trust: M = 3.43, SD = 0.55; Being bullied: M = 3.76, SD = 0.46). Feelings of being bullied during the disability evaluation by the expert were very rare: Only six participants scored below the mean of the scale (i.e. <2.5). There was a strong association between the mean fairness questionnaire score and the Patients' trust score (r = 0.814, p < 0.001, **Fig 2**), as well as between the mean fairness questionnaire score and the Being bullied score (r = 0.556, p < 0.001, **Fig 3**). The factor scores for Rapport and Interviewing skills were considerably associated with the two CPQ scales, whereas the factor scores for Transparency and Case familiarity only showed a weak or no association (**Table 3**).

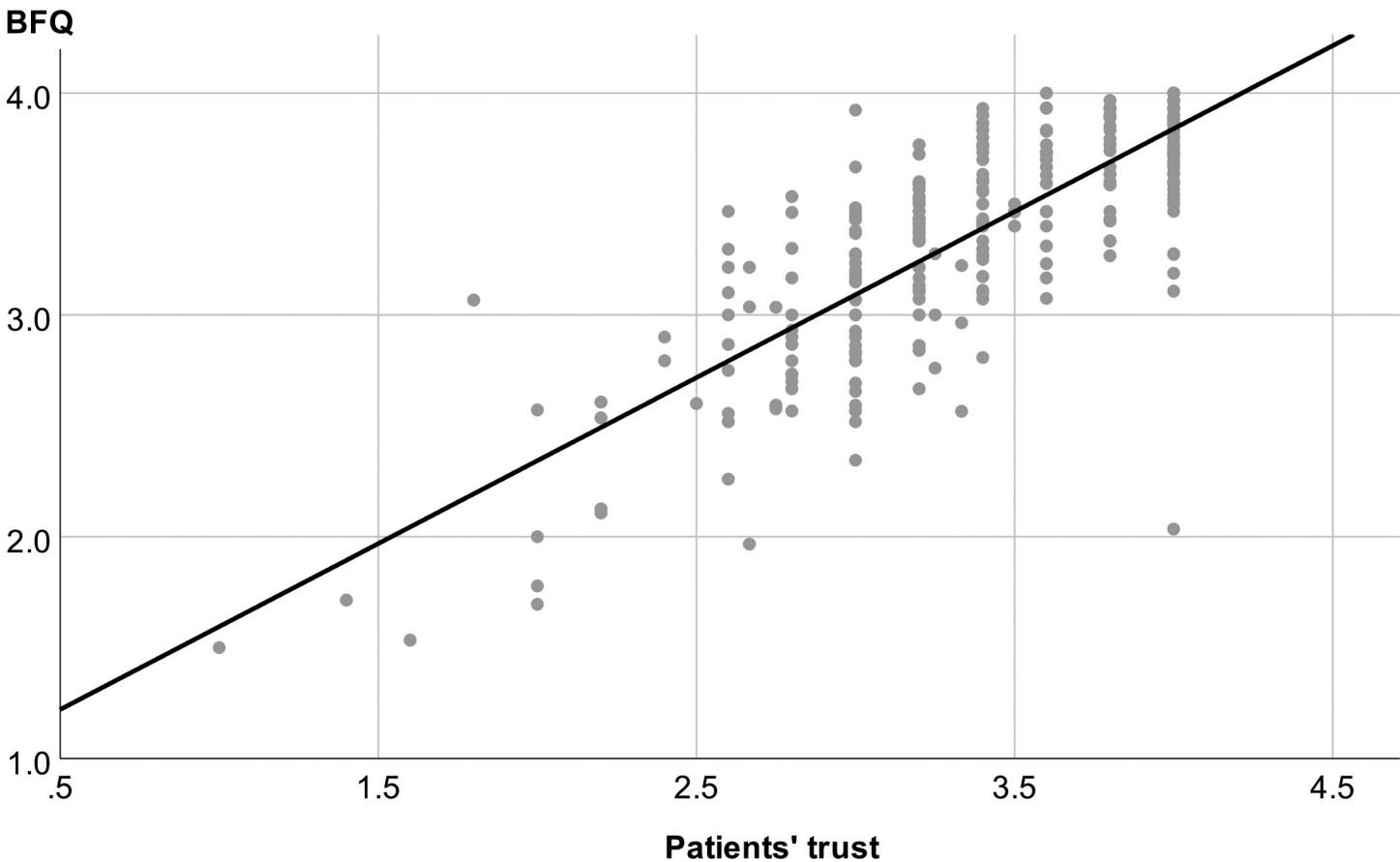

**Fig 2. Convergent validity: Strong positive correlation between the scores of Patients' trust (*CPQ*) and perceived fairness, as quantified by the mean BFQ rating** ($r = 0.814$, $p < 0.001$). Both scorings were strongly left-skewed.

278 participants answered the *SWLS*. The mean rating was 2.51 (SD 0.99), meaning that the participants' life satisfaction ratings were on average below the middle of the scale ("3") and participants tended to be rather dissatisfied with their lives. The mean SWLS ratings showed a small, but significant correlation with the mean BFQ rating (average across all 30 items, $r = 0.238$, $p < 0.001$, **Fig 4**). Claimants with low life satisfaction tended to perceive the evaluation as less fair. The individual factor scores and the SWLS scores showed only weak correlations (**Table 3**).

### Association with other questionnaire ratings and expert rating

The claimants separately rated the perceived fairness of the disability evaluation on a 7-point Likert scale (M = 5.74, SD = 1.24). These ratings showed a strong association with the mean questionnaire score ($r = 0.767$, $p < 0.001$) and small to medium associations with the factor scores between 0.256 and 0.474 (**Table 3**). The corresponding expert ratings ("Do you think the claimant considered the disability evaluation as fair?") were on average quite high (M = 6.09, SD = 0.78) and significantly more positive than the claimant ratings ($F_{1,270} = 18.437$, $p < 0.001$, $\eta_p^2 = 0.064$). The expert ratings showed only a weak association with the mean questionnaire score ($r = 0.186$, **Table 3** and **S1 Fig**).

**BFQ**

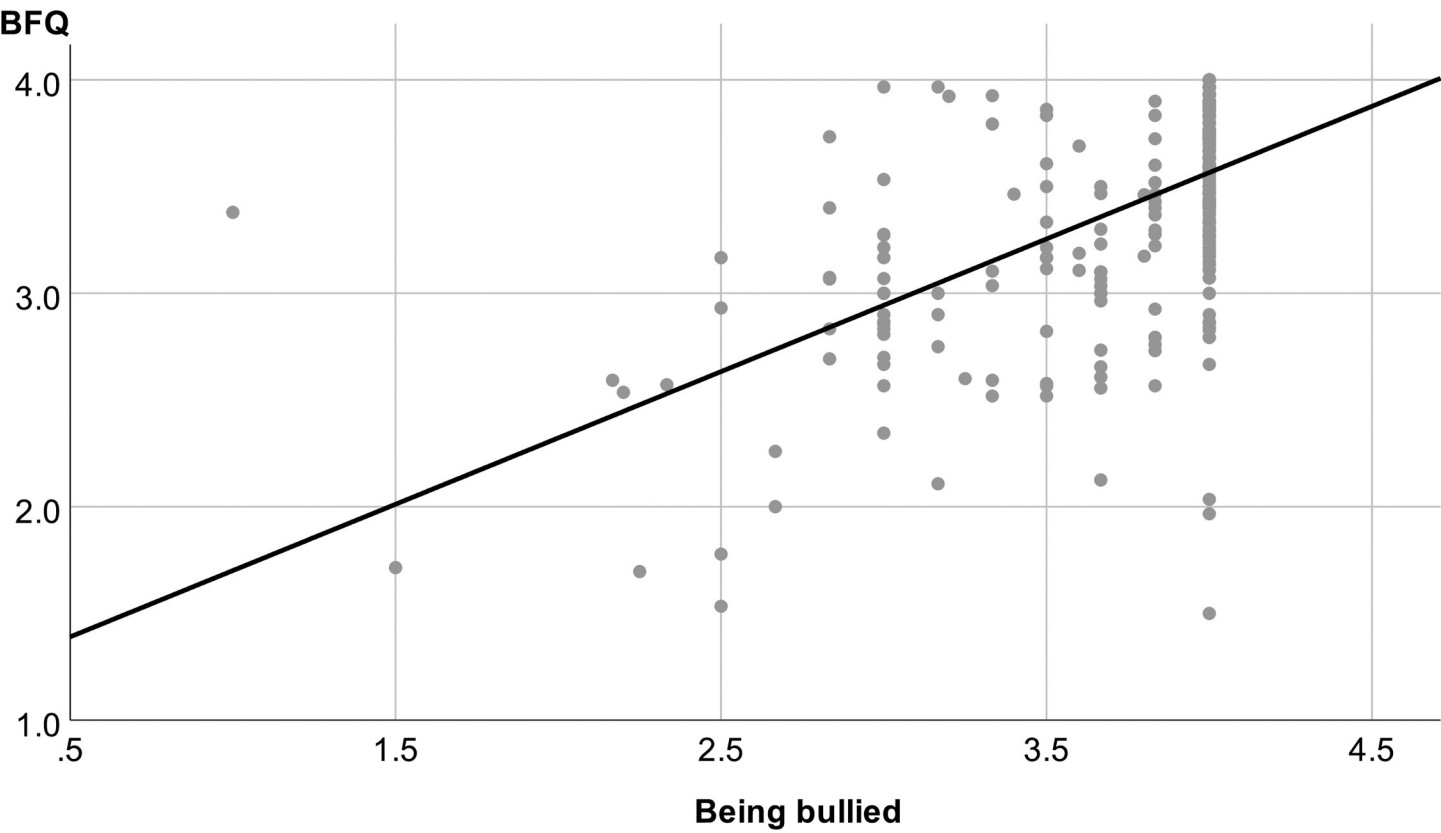

**Fig 3. Convergent validity: Strong positive correlation between the scores of Being bullied (*CPQ*) and perceived fairness, as quantified by the mean BFQ rating (r = 0.556, p < 0.001).** Both scorings were strongly left-skewed.

**Table 3. Basel Fairness Questionnaire (BFQ) correlations.**

| BFQ factor score | CPQ: Patients' trust | CPQ: Being bullied | SWLS: Mean score | BFQ: Patient rating[1] | BFQ: Expert rating[2] |
|---|---|---|---|---|---|
| **Rapport** | 0.490 *** | 0.423 *** | 0.099 | 0.474 *** | 0.094 |
| **Interviewing skills** | 0.601 *** | 0.441 *** | 0.122 * | 0.442 *** | 0.137 * |
| **Transparency** | 0.259 *** | 0.097 | 0.114 | 0.256 *** | 0.107 |
| **Case familiarity** | 0.244 *** | 0.067 | 0.181 ** | 0.342 *** | 0.012 |
| **BFQ mean rating score (30 items)** | 0.814 *** | 0.556 *** | 0.238 *** | 0.767 *** | 0.186 *** |

Correlations between the BFQ scores and the Cologne Patient Questionnaire (CPQ) scores, the Satisfaction with Life Scale (SWLS) score, and the BFQ global ratings

[1] Global rating of the patient ("To what extent do you consider the disability evaluation which has just taken place as fair?"); [2] global rating of the expert ("Do you think the claimant considered the disability evaluation as fair?");

* p < 0.05,

** p < 0.01,

*** p < 0.001

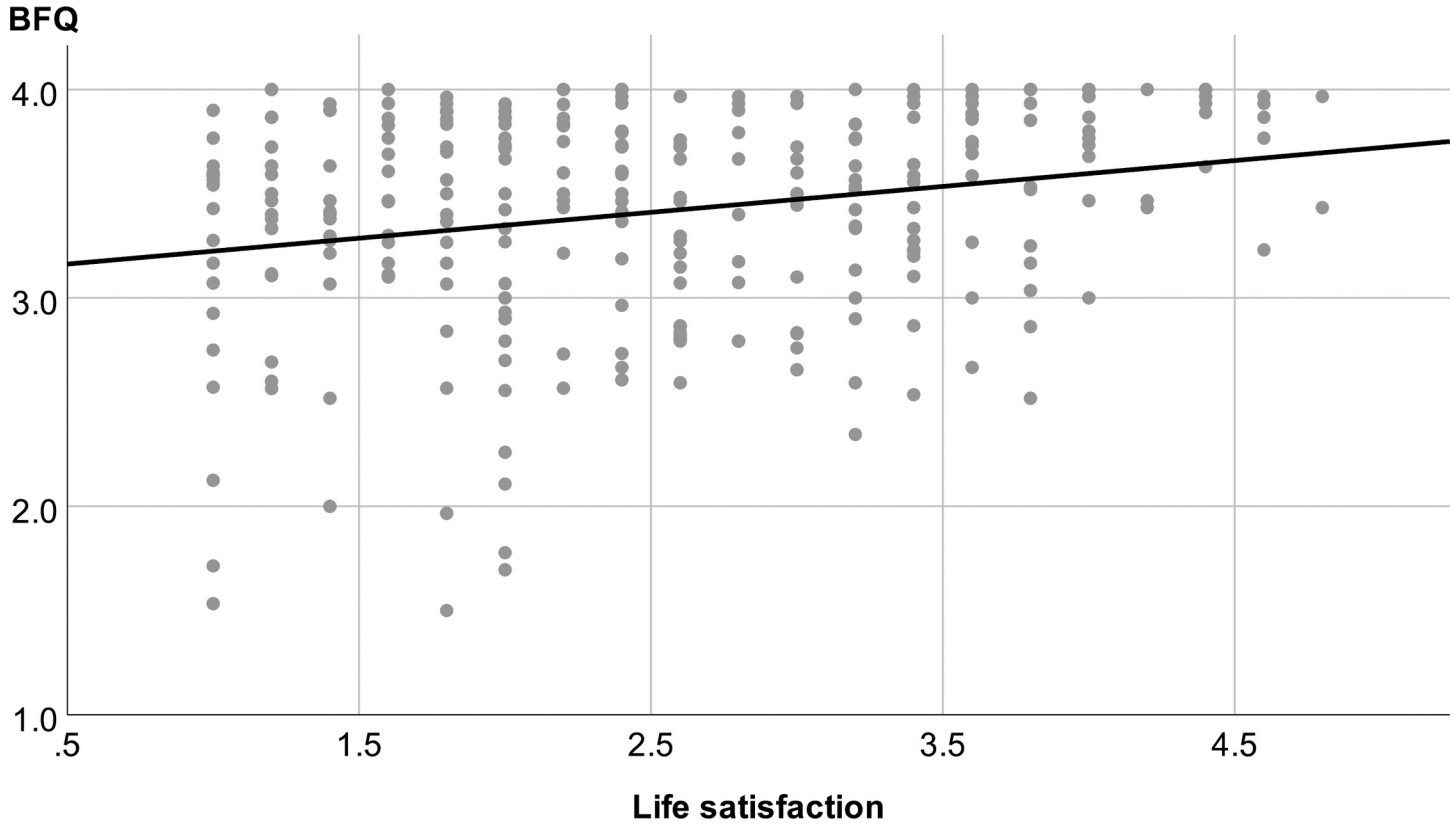

**Fig 4. Divergent validity: The mean BFQ rating was weakly, but systematically modulated by life satisfaction, as measured by the satisfaction with life scale (SWLS, r = 0.238, p < 0.001).** The BFQ scores were strongly left-skewed (most values > 3), whereas the SWLS scores tended to be right-skewed.

## Discussion

In the current study, we aimed to develop and to validate a questionnaire that measures to what degree claimants for disability benefits perceive their evaluation by the medical expert as fair. In the following, we discuss the claimants' mean BFQ ratings on the disability evaluation, the validity of BFQ, the association between the claimant and expert ratings, and the practical implications of the study for quality control in work disability evaluations.

### Mean ratings

The study showed that the claimants' satisfaction with the disability evaluation was on average high, in particular their satisfaction with the atmosphere of trust and respect and with their participation in the interview. Only about 4% of the claimants tended to perceive the evaluation as unfair and rated the disability evaluation below the median of the scale. Only about 2% reported feelings of being bullied by the expert. These data provide a quite positive feedback about the perceived fairness in disability evaluations in Switzerland. Given the lack of instruments, there are currently no defined thresholds for patient satisfaction in disability evaluations. However, experts, assessment centres, and insurance providers would likely consider 2 to 4% of unsatisfied claimants already as best case scenario because claimants often have mental health issues (in forms of personality disorders, depression, or substance abuse, [21]), which might occasionally affect their ratings negatively. Indeed, considering the SWLS data, we found that participants with low life satisfaction were prone to ratings below the median of

the scale, which were virtually absent for participants with good life satisfaction (**Fig 4**). This might suggest that factors like the claimants' life situation, life satisfaction, or personality could have some negative impact on the satisfaction with the disability evaluation. However, the experts need to deal with the claimant's poor life satisfaction, might face an increased difficulty to create and to maintain an atmosphere of mutual trust and respect with these more vulnerable patients, and might ultimately sometimes fail in providing it. Finally, correlation does not imply causation, and it is possible that claimants who experienced the disability evaluation as very negative tended to rate their life satisfaction as more negative.

The very positive ratings across claimants should be interpreted with considerable caution due to potential selection and response bias: First, the four assessment centres that participated in the study represent only a small and regionally restricted sample of Swiss assessment centres. Given this, the four assessment centres might not be representative for (German-speaking) Switzerland. Recruiting study participants via the commissioning disability insurer would ensure a more comprehensive and therefore less biased coverage of assessment centres.

Second, the ethic committee approval covered only the collection of the questionnaire data, which means that no demographic or medical data of participants and non-participants were available, except the demographic data of participants obtained by the questionnaire. This limits the characterization of the study sample, but also prohibits the characterization of non-participants (eligible claimants who were either not invited to participate or who were invited but did not return the questionnaire). Given this, we cannot rule out that the study sample varied from non-responders and was, thus, not representative for claimants with good German command in general. Future studies with the aim to provide quality benchmarks in perceived fairness across Switzerland (or other countries) will require collecting demographic or medical data to ensure representative samples.

Third, the exclusion of claimants with insufficient command of German–albeit necessary by the design of the study–may have introduced additional selection bias. Compared to disability evaluations with native speakers, evaluations requiring an interpreter are more difficult to conduct. Information loss is likely to occur when communication switches from a direct to an indirect format [22]. This information loss might have adverse effects on perceived fairness as well. Such quality deficiencies might be discovered by translated versions of the BFQ, which we plan to use in future studies in order to test whether the perceived fairness varies between work disability evaluations with and without interpreters.

Selection bias might have occurred since study participation was voluntary. While the study achieved a favourable return rate, still, more than 40% of claimants did not return their questionnaire. Nguyen and co-workers [23] have argued that satisfied patients are more likely to return questionnaires than dissatisfied patients. Comparing claimants who completed their questionnaires with those who skipped some responses might suggest that claimants who skipped all questions (i.e., did not return the questionnaire) might have been more negative in their ratings. Even if this assumption was true and non-responders behaved like participants who skipped items, this would have minimally decreased the mean rating of the fairness questionnaire from 3.41 to 3.33.

The high levels of satisfaction might partly reflect a positive skew (the preference of responses towards the favourable end [24]). Such response bias may in parts be related to the asymmetric dyadic interaction in work disability evaluations, with one party evaluating and the other one being evaluated. With these asymmetric social roles in mind, claimants might find it difficult or even risky to provide negative feedback to medical experts, even though anonymity is granted. It is a ubiquitous finding that service recipients report high levels of satisfaction [23]. Nguyen and co-workers therefore argued that the level of satisfaction in absolute terms is often meaningless and instead satisfaction in relative terms should be preferred (i.e.

the comparison of levels of satisfaction across institutions or across time, when assessed by the same instrument).

Finally, the medical experts were aware that they were rated. Although they had no detailed knowledge about the BFQ items, they might have deduced from the expert questionnaire that the quality of the interaction was of importance. Thus, it is possible that medical experts adopted their interviewing behaviour. Such behavioural changes have been conceptualized as *Hawthorne* or observer effect [25]. For ethical reasons, we deliberately discarded the option to keep the medical experts uninformed about the study and took the risk of possible observer effects, when designing the study.

To sum up, the high mean ratings may overestimate the perceived fairness of disability evaluations due to selection bias, response bias, and observer effects. We recommend that in future nationwide surveys, commissioning insurers should distribute the questionnaires and collect the socio-demographic data of responders and non-responders to assure representative samples.

## Validity of the questionnaire

The validation of the questionnaire consisted of three elements: (a) the PCA for assessing construct validity; (b) the association between the questionnaire and CPQ scorings for assessing convergent validity; (c) the association between the questionnaire and SWLS scoring for assessing divergent validity.

**Construct validity.** The PCA revealed four factors, namely (1) Rapport, (2) Interviewing Skills, (3) Transparency, and (4) Case Familiarity. These factors corresponded to the dimensions beforehand identified by subject-matter experts as relevant for the perceived fairness in disability evaluations. Only two items showed factor loadings < 0.5 on any of these four factors. These were the items 12 and 16 (**Table 1**, "The expert asked me how I feel." and "I could ask questions."). The phrasing of both items might be too unspecific to provide valid feedback. Both items will therefore be removed from future versions of the questionnaire.

Albeit the PCA confirms that most items show associations to the dimensions they were designed for, it would be incorrect to consider these dimensions as independent from each other, even though the orthogonal rotation is based on the assumption that the underlying factors (as mathematical reflections of these dimensions) are uncorrelated. First, all items show considerable intercorrelations and all items show an item discrimination > 0.6 (**Table 1**). In addition, the mean rating (across 30 items) and the global rating of the perceived fairness were considerably correlated. This implies that the questionnaire measures *one* quality. This quality ("perceived fairness") can relatively well be quantified by a single rating, as provided with a 7-point Likert scale ("As how fair would you rate the evaluation that has just taken place?"). However, we would argue that it is important to ask for details regarding the perceived fairness because otherwise it would not be possible to infer from a negative feedback ("the disability evaluation was very unfair") to what aspect this negative feedback refers. This would in turn not allow implementing measures against this quality defect. In consequence, the overall BFQ score as well as the four subscores should be reported when reporting information on perceived fairness in disability evaluations.

The perception of the disability evaluation as a whole might affect the evaluation of single dimensions and vice versa ('Halo effect' [26]). Let us assume a claimant leaves the evaluation with a bad feeling, without being able to verbalize what made him feel that way. Being questioned about the various dimensions of the evaluation (like rapport, participation, transparency etc.) might result in negative ratings due to a spillover from his overall negative impression (rather than due to an analytic assessment of the individual dimensions). Reversely, a single annoying aspect (e.g., the claimant felt poorly informed by the expert)

might affect the perceived fairness in general and worsen the rating of other aspects. Individual rating profiles (the assessment in each dimension) need to be interpreted with caution because of such spillover effects.

Nevertheless, rating profiles might become a valuable source of information when averaged across multiple evaluations and compared between experts and assessment centres. Repeated ratings of an expert's interaction by different claimants will minimize the error variance (= variance of unsystematic effects). For example, the expert's mean rating for Transparency might be considerable worse than his ratings for other factors *and* considerably worse than the mean Transparency ratings of other experts. This would indicate that the expert repeatedly lacked transparency, from the perspective of *several* claimants. In the context of quality improvement, the expert would be encouraged to re-think his way of providing information to the claimant.

**Convergent and divergent validity.** The BFQ mean rating was strongly associated with the scores of the two CPQ scales. This indicates that the two instruments measure partly the same underlying dimension and provides evidence for convergent validity of the BFQ. The two scales of the CPQ do not measure independent dimensions; their scores showed a considerable correlation ($r = 0.543$, $p < 0.001$). Patients who felt bullied by the expert did not rate the atmosphere of trust and respect positively. High correlations between the BFQ and CPQ were found for its dimensions Rapport and Interviewing skills, but much less so for Transparency and Case familiarity (**Table 3**). Thus, the latter two dimensions are not covered by the CPQ, as one might infer from the semantic content of the CPQ items.

The BFQ scores and the SWLS scores were only weakly correlated, which provides evidence for divergent validity of the BFQ. A poor rating of the disability evaluation did not reflect poor life satisfaction, even though claimants with poor life satisfaction had some tendency for poorer fairness ratings. Many claimants with poor life satisfaction ($< 3$) gave rather positive fairness ratings ($> 3$, **Fig 4**). The weak association between satisfaction with life and the fairness ratings is also underlined by the differential distribution of the data (with fairness ratings being left-skewed and life satisfaction being right-skewed). This finding provides evidence for divergent validity of the BFQ.

## Expert ratings

The experts poorly predicted how fair the claimants would have perceived the disability evaluation (**S1 Fig**). However, the expert ratings showed only little variance. The experts hardly scored $< 5$, which might be due social desirability (i.e. they do not want to present themselves as unfair.) This small response variance limited the chances to find significant correlations between the patient and expert ratings. Moreover, the patient questionnaire drew the patients' attention to several aspects of the interview (by asking to evaluate them), whereas the expert questionnaire lacked of such details, as it only contained one question referring to perceived fairness. Thus, the weak association between patient and expert ratings might reflect the poor overlap in questionnaire structure and question type. For future studies, the expert questionnaire will be revised, putting a focus on a self-assessment in the four BFQ factor dimensions.

## Implications of the study

Patients' perceived fairness in the context of disability evaluations relates to interactional, informational, and procedural justice. The experts' interaction and their communication skills largely determine to what degree patients perceive the evaluation as fair. By treating patients with respect and by communicating in a clear and comprehensible way, experts can generate the trust that is required for the claimants' participation in the disability evaluation. Patients,

who trust in the experts' way of evaluating their work disability, may be more likely to accept a decision by the insurer even if this decision does not match their original expectations [27].

Fairness, as defined by Goldman and Cropanzano [28], is a *subjective* and moral judgement of a specific situation. For work disability evaluations, the BFQ turns perceived fairness into an indicator that is measurable, quantifiable, and comparable across medical experts and assessment centres. Such an indicator would be a crucial component of quality control. It needs to be stressed that claimants may perceive their disability evaluation as fair, but the expert assessment might nonetheless suffer from other quality deficiencies, such as missing and incomplete information, false information, or misattributions. To avoid quality deficiencies, reports should be standardized, transparent, and based on an objective, valid, and (across claimants) comparable assessment, as requested for injury-related disability evaluations [29]. For improving the fairness of work disability evaluations, the quality control of perceived fairness in itself is not sufficient and needs to be complemented by other measures, such as the implementation of a structured functional evaluation process [30], as well as the standardization of the evaluation process [3], accompanied by appropriate training [31].

## Conclusions

We developed the first questionnaire that measures to what degree patients perceive their disability evaluation as fair. The BFQ has good internal consistency, construct validity, convergent and divergent validity. The questionnaire has great potential for quality evaluation and improvement purposes in disability evaluations. However, selection bias, response bias and observer effects may have influenced the currently observed ratings and thereby provided a euphemistic image of the quality of the disability evaluations in Switzerland. By using strategies that counter selection bias, cover response bias, and tackle observer effects, future studies should be able to describe the quality of disability evaluations and to discover potential systematic quality deficiencies more accurately.

## Supporting information

**S1 Fig. Associaation between BFQ mean ratings and the expert rating.**
(PDF)

**S1 Table. Rotated factor solution (2 components extracted).**
(DOCX)

## Acknowledgments

We greatly appreciated the contribution of our subject-matter experts of insurance medicine (Dr. Ulrike Hoffmann-Richter, Dr. Joerg Jeger, Dr. Renato Marelli, and Dr. Gregor Risi) in item selection, formulation, and evaluation. Three of them (JJ, RM, GR) also contributed to clustering the items according to their content. We thank the Dutch Employee Insurance Agency (UWV) for sharing the Dutch questionnaire on patient satisfaction with the expert communication. We greatly valued the support of our study by the medical assessment centres (MEDAS Lucerne, ZMB Basel, BEGAZ Binningen, asim Basel) for the enrolment of study participants. We thank Dr. Baerbel Brigger for pre-testing our questionnaire and Nicole Vogel for her contribution in data collection and study organization.

## Author Contributions

**Conceptualization:** Katrin Fischer, Regina Kunz.

**Data curation:** Regine Lohss.

**Formal analysis:** Timm Rosburg.

**Funding acquisition:** Regina Kunz.

**Methodology:** Timm Rosburg.

**Project administration:** Regine Lohss, Brigitte Walter Meyer.

**Supervision:** Wout de Boer, Regina Kunz.

**Writing – original draft:** Regine Lohss, Timm Rosburg.

**Writing – review & editing:** Regine Lohss, Timm Rosburg, Monica Bachmann, Brigitte Walter Meyer, Wout de Boer, Katrin Fischer, Regina Kunz.

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
