## [Decision Letter · Decision Letter 0]

19 May 2020

PONE-D-20-09205

Perceived fairness of claimants undergoing a work disability evaluation: Development and validation of the Basel Fairness Questionnaire

PLOS ONE

Dear Dr. Rosburg,

Thank you for submitting your manuscript to PLOS ONE. After careful consideration, we feel that it has merit but does not fully meet PLOS ONE’s publication criteria as it currently stands. Therefore, we invite you to submit a revised version of the manuscript that addresses the points raised during the review process. Please ensure that you respond to each point raised below, and in the attachment uploaded by Reviewer 1, in your revision.

We would appreciate receiving your revised manuscript by Jul 03 2020 11:59PM. To enhance the reproducibility of your results, we recommend that if applicable you deposit your laboratory protocols in protocols.io, where a protocol can be assigned its own identifier (DOI) such that it can be cited independently in the future. For instructions see: http://journals.plos.org/plosone/s/submission-guidelines#loc-laboratory-protocols

We look forward to receiving your revised manuscript.

Kind regards,

Melita J. Giummarra

Academic Editor

PLOS ONE

Journal Requirements:

'The study was funded by the Swiss National Accident Insurance Fund, Suva, Lucerne, granted to RK. The funder had no role in study design, data collection and analysis, decision to publish, or preparation of the manuscript.'

We note that you received funding from a commercial source: Swiss National Accident Insurance Fund

Reviewers' comments:

Reviewer's Responses to Questions

**Comments to the Author**

1. Is the manuscript technically sound, and do the data support the conclusions?

Reviewer #1: Partly

Reviewer #2: Yes

2. Has the statistical analysis been performed appropriately and rigorously? 

Reviewer #1: Yes

Reviewer #2: N/A

3. Have the authors made all data underlying the findings in their manuscript fully available?

Reviewer #1: No

Reviewer #2: Yes

4. Is the manuscript presented in an intelligible fashion and written in standard English?

Reviewer #1: Yes

Reviewer #2: Yes

5. Review Comments to the Author

Reviewer #1: The authors developed and validated a comprehensive questionnaire aimed at measuring the perceived fairness of disability claimants that underwent a medical work capacity evaluation. The study covers an important and under-researched topic in disability evaluation and approaches recent criticism that have been raised, in particular with regard to the neglect of the claimants view. The questionnaire appears particularly promising as a basis for quality improvement purposes in the evaluation process.

Overall, the paper is interesting and well written and can be accepted pending appropriate revision. There are slight improvement opportunities regarding the English language and a less pedestrian and more direct wording. In addition, the study suffers from an overload of research steps and corresponding detail information presented in the results section. I would suggest thinking about dropping the last two methodological steps (comparison with expert rating, preceding questionnaires) which do not add sufficient robust information on the questionnaire's external validity (or alternatively I suggest reflecting on this in the study limitations section). Moreover, the authors should add a description of the sample characteristics and carefully check that all steps addressed in the results and discussion section are outlined in the methods (i.e. primarily comparison to claimant's preceding evaluations). For suggestions that are more detailed please see attachment.

Reviewer #2: The paper presents novel research that aims to measure the claimant experience, specifically perceived fairness, of disability evaluations. The research has been well justified, and the design and methodology used seemed appropriate. The results well explained and outlined in detail.

I had only a few queries about the development of the questionnaire and recruitment of participants and medical expert role. There is little detail explanation about the "subject-matter experts", and how they influenced the dimensions of the questionnaire. On the questionnaire, the principles of informational and interactive justice can easily be recognised in the questions, yet it is difficult to see how procedural justice is measured. It may be helpful for international readers to give a brief explanation of the Swiss disability evaluation process and note whether it is mandatory for claimants to attend a disability assessment. Might mandatory attendance influence the claimant to rate the evaluation positively if they feared a negative outcome?

It is unclear how the assessment centres or medical experts were chosen or enrolled in the study. Also how soon after examining a claimant did the medical experts complete their ratings? Were the medical experts aware the claimants were rating their performance? Why were 12 claimants not rated?

Overall the limitations of the research, and potential mechanisms for redress were well considered and thoroughly discussed. The recommendations for further study and evaluation of the questionnaire are also logical and commendable. With the further testing and refinement proposed by the authors, it seems this questionnaire could be used as a valuable feedback mechanism to inform and improve medical expert performance and thus the quality of assessment. This should then also benefit claimant experience. Creation of a tool to measure the fairness of disability evaluations is an important contribution to the field and potentially has applicability across jurisdictions.

6. PLOS authors have the option to publish the peer review history of their article (what does this mean?). If published, this will include your full peer review and any attached files.

Reviewer #1: No

Reviewer #2: No

---

## [Decision Letter · Decision Letter 1]

7 Aug 2020

PONE-D-20-09205R1

Perceived fairness of claimants undergoing a work disability evaluation: Development and validation of the Basel Fairness Questionnaire

PLOS ONE

Dear Dr. Rosburg,

Thank you for submitting your manuscript to PLOS ONE. After careful consideration, we feel that it has merit but does not fully meet PLOS ONE’s publication criteria as it currently stands. Therefore, we invite you to submit a revised version of the manuscript that addresses the points raised during the review process. I picked up some minor typographical corrections in your manuscript that should be addressed before I can accept it for publication. The suggested corrections are noted on a PDF of your revision.

We look forward to receiving your revised manuscript.

Kind regards,

Melita J. Giummarra

Academic Editor

PLOS ONE

Reviewers' comments:

Reviewer's Responses to Questions

**Comments to the Author**

1. If the authors have adequately addressed your comments raised in a previous round of review and you feel that this manuscript is now acceptable for publication, you may indicate that here to bypass the “Comments to the Author” section, enter your conflict of interest statement in the “Confidential to Editor” section, and submit your "Accept" recommendation.

Reviewer #1: All comments have been addressed

Reviewer #2: All comments have been addressed

2. Is the manuscript technically sound, and do the data support the conclusions?

Reviewer #1: Yes

Reviewer #2: (No Response)

3. Has the statistical analysis been performed appropriately and rigorously? 

Reviewer #1: Yes

Reviewer #2: (No Response)

4. Have the authors made all data underlying the findings in their manuscript fully available?

Reviewer #1: Yes

Reviewer #2: (No Response)

5. Is the manuscript presented in an intelligible fashion and written in standard English?

Reviewer #1: Yes

Reviewer #2: (No Response)

6. Review Comments to the Author

Reviewer #1: I would like to thank the authors for their careful revision of the manuscript and for providing a satisfactory response to all issues raised by the reviewers. The study represents an important step towards increased fairness in disability evaluation and opens up an area with great potential.

Reviewer #2: (No Response)

7. PLOS authors have the option to publish the peer review history of their article (what does this mean?). If published, this will include your full peer review and any attached files.

Reviewer #1: No

Reviewer #2: No

---

## [Editor Report · Decision Letter 2]

27 Aug 2020

Perceived fairness of claimants undergoing a work disability evaluation: Development and validation of the Basel Fairness Questionnaire

PONE-D-20-09205R2

Dear Dr. Rosburg,

We’re pleased to inform you that your manuscript has been judged scientifically suitable for publication and will be formally accepted for publication once it meets all outstanding technical requirements.

Kind regards,

Melita J. Giummarra

Academic Editor

PLOS ONE
---

## [Editor Report · Acceptance letter]

4 Sep 2020

PONE-D-20-09205R2 

Perceived fairness of claimants undergoing a work disability evaluation: Development and validation of the Basel Fairness Questionnaire 

Dear Dr. Rosburg:

I'm pleased to inform you that your manuscript has been deemed suitable for publication in PLOS ONE. Congratulations! Your manuscript is now with our production department. 

Kind regards, 

on behalf of

Dr. Melita J. Giummarra 

Academic Editor

PLOS ONE